# High Manganese Content of Lipid NanoMn (LNM) by Microfluidic Technology for Enhancing Anti-Tumor Immunity

**DOI:** 10.3390/pharmaceutics16040556

**Published:** 2024-04-19

**Authors:** Jiawei Sun, Jingjing Gong, Lidong Gong, Chuanda Zhu, Longhao Li-Yang, Jingya Wang, Yuanyuan Yang, Shiming Zhang, Silu Liu, Ji-Jun Fu, Pengcheng Xu

**Affiliations:** 1Department of Pharmaceutics, College of Pharmacy, Inner Mongolia Medical University, Hohhot 010110, China; 2021110078@stu.immu.edu.cn (J.S.); 2021110077@stu.immu.edu.cn (S.L.); 2Beijing Key Laboratory of Tumor Systems Biology, Institute of Systems Biomedicine, School of Basic Medical Sciences, Peking University Health Science Center, Beijing 100191, China; gongjingjing@bjmu.edu.cn (J.G.); gld.pku@bjmu.edu.cn (L.G.); 2211110006@pku.edu.cn (C.Z.); 1910305210@pku.edu.cn (L.L.-Y.); 2311110083@stu.pku.edu.cn (J.W.); 1610305210@pku.edu.cn (Y.Y.); 2111110690@bjmu.edu.cn (S.Z.); 3School of Pharmaceutical Sciences, Guangzhou Medical University, Guangzhou 511436, China

**Keywords:** microfluidics, manganese, mass production, cGAS-STING, activation of immunity

## Abstract

Immunotherapy is a clinically effective method for treating tumors. Manganese can activate the cGAS-STING signaling pathway and induce an anti-tumor immune response. However, its efficacy is hindered by non-specific distribution and low uptake rates. In this study, we employed microfluidic technology to design and develop an innovative preparation process, resulting in the creation of a novel manganese lipid nanoparticle (LNM). The lipid manganese nanoparticle produced in this process boasts a high manganese payload, excellent stability, the capacity for large-scale production, and high batch repeatability. LNM has effectively demonstrated the ability to activate the cGAS-STING signaling pathway, induce the production of pro-inflammatory cytokines, and inhibit tumor development. Notably, LNM does not require combination chemotherapy drugs or other immune activators. Therefore, LNM presents a safe, straightforward, and efficient strategy for anti-tumor immune activation, with the potential for scalable production.

## 1. Introduction

Immunotherapy is a crucial approach for treating tumors, capable of stimulating and enhancing the body’s immune response, which in turn, inhibits and eliminates tumor cells [1]. Metal ions, especially manganese ions (Mn^2+^), play a significant role in anti-tumor immunity by activating the cyclic GMP-AMP synthase stimulator of the interferon gene (cGAS-STING) signaling pathway [2]. This activation facilitates dendritic cell (DC) maturation and migration while enhancing the cytotoxicity of killer T cells (CTLs) and natural killer cells (NK cells) [3].

Manganese ions are limited since it is difficult for them to be absorbed by cells, and they have non-specific distribution. Recently, to overcome the limitations of free Mn^2+^, there have been promising developments in Mn-based lipid nanoparticles due to their simple components and strong biocompatibility. At present, lipid nanoparticles (LNPs) are one of the important technologies in lipid carrier drug delivery systems, but the mass production of nanoscale manganese continues to present considerable challenges. Additionally, previous research has shown that manganese is often used solely as an immune adjuvant to activate host immunity, usually in combination with chemotherapy drugs or other immune activators to achieve a highly effective anti-tumor response [4]. While nanomanganese demonstrates considerable potential in cancer immunotherapy, substantial limitations persist in both its production and application [5]. Consequently, the development of an efficient, stable, and reproducible production process remains a critical obstacle that must be addressed for successful translation into clinical practice.

Microfluidic technology is a widely adopted technique for the production of liposomes [6]. Recent studies demonstrated the successful preparation of cationic liposomes with high transfection efficiency and excellent reproducibility through microfluidic control, employing various mixer structures and flow rate ratios [7]. In this study, we harnessed microfluidic technology to develop a novel variant of manganese lipid nanoparticles, referred to as LNM. Importantly, our assessments confirm that LNM can effectively activate the cGAS-STING pathway and elicit an effect without the need for combining it with chemotherapy drugs or other immune activators. These findings open up the potential for the further clinical translation of manganese nanoparticles.

## 2. Materials and Methods

### 2.1. Materials

MnCl_2_·4H_2_O, Na_2_HPO_4_·12H_2_O, waterless ethanol, dichloromethane (CH_2_Cl_2_), and cyclohexane were purchased from Tong Guang (Beijing, China). Phospholipids (DOTAP, cholesterol, and DSPE-PEG 2000) were ordered from AVT Pharmaceutical Tech Co., Ltd. (Shanghai, China). DOPA and IGEPAL^®^ CO-520 were purchased from Sigma-Aldrich (Shanghai Branch, Shanghai, China). CCK-8 was purchased from TargetMol, USA. Mouse anti-GAPDH monoclonal antibody, horseradish peroxidase (HRP)-labeled goat anti-rabbit IgG (IgG-HRP), HRP-labeled goat anti-mouse IgG (IgG-HRP) were purchased from Invitrogen (Beijing agent, Beijing, China). Rabbit anti-STING monoclonal antibody was purchased from Solarbio Science&Technology Co., Ltd. (Beijing, China). Rabbit anti-P-STING monoclonal antibody was purchased from Biodragon (Beijing, China). Rabbit anti-P-IRF3 monoclonal antibody was purchased from Biosynthesis Biotechnology Co., Ltd. (Beijing, China). Mouse IFN-γ Precoated ELISA Kit, mouse IL-6 Precoated ELISA Kit were obtained from Dakewe Biotech Co., Ltd. (Shenzhen, China). The fluorescence antibodies (FITC-anti-CD 3, Brilliant Violet-anti-CD 4, and APC-anti-CD 8a) were purchased from BioLegend, (Beijing agent, Beijing, China). Deionized water was used in this work. Other solvents and reagents were of analytical grade without further purification before use.

The Jurkat, MC38, 3T3, and HEF cell lines used in this study were derived from our laboratory’s cell bank and were passaged no more than 10 times.

Sexually matured (over 4-week-old female) C57BL/6N were purchased from Nonc Biotech Co., Ltd. (Beijing, China). Mice were kept under barrier housing facilities with controlled temperature (24–26 °C) and light (12 h light and 12 h dark cycles), with food and water ad libitum. Briefly, 16 female C57BL/6N mice were prepared. All of the animal protocols related to this study were reviewed and approved by the Institutional Animal Care and Use Committee of the Peking University Health Science Center.

### 2.2. Preparation of Liposomes Using Microfluidics

The preparation process of LNM is illustrated in Figure 1. To achieve this, we utilized a single-channel syringe pump to introduce a 500 mM MnCl_2_ solution into the central channel of the microfluidic chip. Simultaneously, we injected a mixed oil phase composed of CO-520 and cyclohexane into the adjacent channel, resulting in the formation of a Mn^2+^ microemulsion. To further enhance the process, we reintroduced Na_2_HPO_4_ (25 mM) and the mixed oil phase at the same flow rate, with the addition of DOPA (23 mg/mL) into the phosphoric acid phase. The flow rate played a critical role in this step. We conducted a screening of LNM cores at different flow rate ratios (FRR) and identified the optimal flow rate, as detailed in Table 1. To finalize the core preparation, the oil phase was separated using absolute ethanol and high-speed centrifugation (10,000× *g*, 20 min) to obtain the desired nanoprecipitate. After collecting the nanoprecipitate, we mixed 3 mM DSPE-PEG2000 with 10 mM DOTAP/cholesterol (1:1). This mixture was used to form a lipid film through the thin film hydration method. Subsequently, we dispersed the lipid film into 800 μL of ultrapure water using ultrasonic methods, resulting in the formation of a milky white solution.

The LNM nanoparticles were diluted 100-fold with distilled water using the PSS ZPW388-NICOMP particle quantification system. Particle size was determined using the dynamic light scattering (DLS) method. The LNM nanoparticles were diluted 10-fold with water before being deposited onto a 200-mesh carbon-coated grid. After the LNM nanoparticles adhered to the carbon grid, excess liquid was removed using absorbent paper. Subsequently, the grid was negatively stained with uranyl acetate solution, excess staining solution was removed, and the carbon grid was air-dried under an infrared lamp. The morphology of LNM was observed using transmission electron microscopy (TEM, JEOL174 1200EX).

The LNM nanoparticles were diluted 10-fold with water. Subsequently, the diluted LNM nanoparticles were then vacuum-treated on a 200-mesh carbon support grid using a plasma cleaner. Following this, the diluted LNM nanoparticles were dropped onto the carbon grid and air-dried under an infrared lamp. The elemental composition of LNM was observed using scanning transmission electron microscopy (STEM, FEI Tecnai F30).

### 2.3. LNM Activates cGAS-STING Pathway

Western blot analysis was employed to determine whether LNM activated the cGAS-STING pathway in cells. Jurkat cells, with an initial seeding density of 1 × 10^5^ cells per well, were cultivated in 6-well plates. These cells were then exposed to LNM at various concentrations (10 μM, 20 μM, 50 μM, and 100 μM) for a 14 h incubation period. Subsequently, the cells were harvested and lysed. Protein samples were separated using SDS-PAGE and then transferred onto PVDF membranes. To prevent non-specific binding, the hydrophobic binding sites on the PVDF membrane were blocked with a 5% BSA solution. The PVDF membrane was then incubated with the primary antibodies specific to the target proteins. Subsequent incubation with goat anti-rabbit IgG labeled with horseradish peroxidase (HRP) enabled the visualization of STING, P-STING, and P-IPF3 protein expressions.

### 2.4. Cytotoxicity Assay for LNM

Cell Counting Kit-8 was used to detect the cytotoxicity of LNM. 3T3 cells and HEF cells were individually seeded into separate 96-well plates. Various concentrations of MNMn (5 μM, 10 μM, 15 μM, 20 μM, 25 μM, 30 μM, 40 μM, 50 μM, and 100 μM) were introduced, and the cells were incubated for a 24 h duration. Following incubation, the original medium was discarded, and a fresh medium, along with the CCK-8 solution, was added. The cells were then incubated in the dark at 37 °C for 30 min. The absorbance at 450 nm was subsequently measured using a microplate reader.

### 2.5. Determination of Intracellular Manganese Content

Mn^2+^ content in MC38 cells was determined using inductively coupled plasma mass spectrometry (ICP-MS). MC38 cells underwent treatment with either 10 μM MnCl2 or LNM for specific durations (0 h, 2 h, 4 h, 6 h, 8 h, 10 h, 12 h, and 24 h). Subsequently, the harvested and processed cells were subjected to overnight cold digestion with 0.1 mL of concentrated nitric acid added per tube. Afterward, 0.075 mL of Rh (20 ng/mL) internal standard solution was added, followed by dilution to 1.5 mL with distilled water. Intracellular Mn^2+^ content was quantified using ICP-MS analysis.

### 2.6. Cell Uptake Was Detected Using Flow Cytometry

DSPE-PEG2000 was labeled with NHS-Cy5, and LNM was prepared. MC38 cells underwent treatment with 10 μM of NHS-Cy5-labeled LNM for specified durations (0 h, 2 h, 4 h, 6 h, 8 h, and 10 h). Subsequently, the treated cells were collected and subjected to analysis using flow cytometry.

### 2.7. LNM Distribution Experiment

C57BL/6N female mice were randomly divided into two groups (*n* = 2) and intraperitoneally injected with 200 μL of solution containing the same concentration of free DiR or LNM-DiR. Within 24 h, real-time fluorescence imaging of the mice was performed using a mouse in vivo imaging system. The distribution of the formulations in the mice was observed at 1 h, 2 h, 4 h, 8 h, and 24 h. After 24 h, the mice were euthanized, and organs such as the heart, liver, spleen, lungs, kidneys, and tumors were collected. Fluorescence imaging of the organs was conducted using a fluorescence imaging system.

### 2.8. LNM Activates Immunity

C57BL/6N MC38 tumor-bearing mice (female, 4–8 weeks old) were randomly assigned to three groups (*n* = 4). On days 0, 2, 4, 6, 8, 10, 12, and 14, the mice in these groups were intraperitoneally administered with either 9% physiological saline, MnCl_2_, or LNM. On day 16, the mice were euthanized. Tumors and spleens were collected for photography and weight measurements. Serum and spleen cells were also collected. The serum was used for the detection of immune-related cytokines (IFN-γ, IL-6) and blood biochemical safety, while spleen cells were subjected to flow cytometry for antibody detection. All of the animal protocols related to this study were reviewed and approved by the Institutional Animal Care and Use Committee of the Peking University Health Science Center.

### 2.9. Serum Cytokine Levels of Immunized Mice Were Detected by ELISA

The expression of IFN-γ and IL-6 cytokines in serum was quantified using an ELISA kit. To summarize, standard samples were gradient-diluted, and serum from the four groups of treated mice was applied to pre-coated strips. Subsequently, the detection antibody was added to facilitate a 90 min incubation at 37 °C, allowing binding between the antigen and antibody in the samples. Following this, a washing solution was used to remove any unbound substances, and the enzyme-labeled antibody was introduced for secondary coupling and a 30 min incubation at 37 °C. After additional washing, the reaction was developed using tetramethylbenzidine (TMB), and the reaction was halted after 20 min. The optical density was then read at 450 nm using a microplate reader. Before statistical analysis, the data were normalized and equilibrated using log_2_ endpoint titers.

### 2.10. Detection of Mouse Spleen Cells by Flow Cytometry

In summary, following 16 days of administration, the mice were euthanized, and their spleens were isolated. Non-target tissues were excised, and the spleen tissues were then diced into small pieces in pre-cooled PBS. These tissue fragments were homogenized using a 2 mL syringe plunger and passed through a 100 μm cell filter to obtain a suspension of spleen cells. After centrifugation, red blood cell lysate was introduced and incubated on ice for 5 min, followed by another round of centrifugation. The cells were subsequently resuspended and counted. Appropriate fluorescent antibodies (CD3, CD4, CD8a) were added and allowed to incubate for 30 min in the dark. The cells were washed and resuspended once in PBS, and the relative content of the target antibodies was measured using flow cytometry.

### 2.11. In Vivo Safety of LNM

In order to evaluate the safety of LNM in vivo, serum from the mice in each group was collected on the 16th day of administration. The levels of liver function indicators (ALT, AST, and ALP) and renal function markers (UA, CREA-J, and UREA) in the serum of the mice were analyzed using an automated animal biochemical analyzer. The main organs of the mice, including the heart, liver, spleen, lung, and kidney, were harvested to prepare frozen sections. Hematoxylin and eosin (HE) staining was performed to examine the histological changes in the organs and tissues of each group.

### 2.12. Data Processing

Data were processed by GraphPad Prism 8.0.1 (GraphPad Software Inc., La Jolla, CA, USA), and results were expressed as mean ± standard deviation (SD). Statistical analysis was considered statistically significant using one-way analysis of variance (ANOVA) or Student’s *t*-test (*p* < 0.05), expressed as follows: ns, no significance; *, *p* < 0.05; **, *p* < 0.01; ***, *p* < 0.001; ****, *p* < 0.0001).

## 3. Results

### 3.1. Process Screening of LNM Nanoparticles

Microfluidic techniques were employed to design and construct LNM lipid nanoparticles. These nanoparticles consist of a core composed of manganese phosphate microprecipitation, enveloped by a phospholipid bilayer (Figure 1). Overall, LNM is a nanoparticle encapsulated by a manganese ion core and an outer layer of phospholipids, while the manganese ion core specifically refers to the microprecipitation of manganese hydrogen phosphate. In the preparation process, CO-520 surfactant is used to form a mixed oil phase with cyclohexane, which helps maintain the strong stability of the subsequent Mn^2+^ microemulsion and HPO_4_^2−^ microemulsion. DOPA anionic lipid material is incorporated into the HPO_4_^2−^ microemulsion, followed by mixing with Mn^2+^ microemulsion. This process allows the hydrophilic end of DOPA, containing phosphate, to be inserted into the precipitate, while the hydrophobic long hydrocarbon chain extends into the oil phase, resulting in a core with a single layer of phospholipids. Subsequently, the cationic lipid material DOTAP is used to wrap the entire core, and a stable phospholipid bilayer will be formed due to electrostatic action, which will not lead to the formation of large aggregates of nanoparticles. The reason why no shear force is introduced in the last step of forming the phospholipid bilayer is that the core of manganese biphosphate has a certain weight, like a pebble, and because gravity cannot be uniformly dispersed in the solution, it cannot be uniformly introduced into the microfluidic channel, so we use the thin film hydration method to add the second layer of phospholipids.

Throughout the process exploration, key factors affecting the properties of LNM nanoparticles were identified. These factors included the core particle size of the microprecipitation, the ratio of microprecipitation to the inner phospholipid, and the ratio of the core to the outer phospholipid within the monolayer phospholipid [8,9].

#### 3.1.1. The Effect of Flow Rate on the Particle Size of LNM Core

The high shear mixing method is a batch process utilized for the preparation of LNM. This method entails the sequential creation of a metal ion microemulsion, a phosphate phase microemulsion, and the formation of LNM cores. Ensuring the formation of stable microemulsions necessitates the simultaneous injection of the aqueous and organic phases, making the flow rate ratio (FRR) between these phases a pivotal factor [10] in achieving this stability. To investigate how the flow rate of the two-phase system affects the size of the LNM core, we designed an experiment. In Table 1, we illustrate our approach, wherein we maintained a constant volume ratio of the water phase to the oil phase but adjusted the flow rate, ultimately yielding 24 distinct samples. These samples included particles with monolayer phospholipid cores and their corresponding complete LNM nanoparticles. We then diluted and deposited these samples onto 200-mesh carbon films to prepare TEM samples. Subsequently, we employed transmission electron microscopy to observe the morphology of LNM monolayer phospholipid cores and complete nanoparticles produced at varying flow rates, counting their respective particle sizes. In instances where the particle size is too small to support the phospholipid bilayer, it leads to the failure of LNM nanoparticle formation [11]. Conversely, when the particle size is too large, LNM nanoparticles tend to aggregate and precipitate. As a result, we determined that an FRR of 0.2 was the optimal flow rate, as dictated by the observed particle size and morphology.

#### 3.1.2. The Effect of the Ratio of Microprecipitates in the Inner Phospholipid on the Dispersion and Uniformity of the LNM Core

Once the optimal flow rate ratio was established, we proceeded to investigate the impact of varying the quantity of phospholipids added to the inner layer on LNM nanoparticles. To this end, we created six distinct single-layer phospholipid microprecipitates for LNM by incorporating varying amounts of DOPA, as outlined in Table 2. We examined the particle size of these microdeposits. Notably, when DOPA doses are too high or too low, the uniformity and dispersion of LNM are adversely affected. Following this evaluation, we determined that a DOPA dosage of 23 mg/mL was the optimal concentration for our purposes.

#### 3.1.3. The Effect of the Ratio of the LNM Core to the Outer Phospholipid on the Load Ratio of Mn

By controlling the flow rate and the amount of phospholipid added in the inner layer, we determined the optimal technological conditions for the production of LNM using morphology. However, the actual active component of LNM is manganese in its core, and the ratio of the outer phospholipid to the single-layer phospholipid core can significantly affect the stability and uniformity of nanoparticles, thus affecting the loading rate of effective Mn. Therefore, we obtained 10 kinds of liposome samples by changing the amount of phospholipid in the core and outer layer. As shown in Table 3, the manganese content of the samples was analyzed using inductively coupled mass spectrometry (ICP-MS), and the particle size was statistically analyzed. Finally, the 10# LNM nanoparticles with the highest Mn content, moderate particle size and best dispersion were selected.

### 3.2. The Characterization of LNM

After the above screening process, we finally determined that the flow rate ratio is equal to the ratio of precipitated core to inner phospholipid, and the ratio of the single phospholipid core to the outer phospholipid core offers the best conditions for processing. The LNM lipid nanoparticles prepared using this process were fully characterized. As shown in Figure 2A, under transmission electron microscopy (TEM), it can be seen that LNM is a spherical nanoparticle composed of a microprecipitated core and outer layer lipid, with a uniform size and good dispersion, and a particle size of approximately 30–60 nm. Further energy spectrum scanning analysis showed that LNM nanoparticles were effectively loaded with manganese ions (Figure 2B). In order to further investigate the stability of the process and the repeatability of LNM nanoparticles between batches, we randomly selected five time points over the course of 6 months to repeatedly prepare LNM particles five times, and then measured their particle size on the preparation day [12]. As shown in Figure 2C, based on the results of dynamic light scattering (DLS), the diameter difference of nanoparticles prepared in different batches is less than 10 nm, and the diameter is approximately 60 nm, indicating that LNM has good batch repeatability. The size difference in the TEM images can be attributed to the hydrated particle size of the nanoparticles measured using DLS [13]. Next, we further investigated the long-term placement stability of LNM nanoparticles. As shown in Figure 2D,E, LNM nanoparticles placed in PBS or PBS containing 5% serum can maintain their particle size balance for an extended period at both 4 °C and 37 °C, with only a slight increase in particle size.

### 3.3. The Verification of LNM Uptake and Immune Activation Ability

The bioactivity of LNM nanoparticles hinges on their complete internalization within cells [14,15]. To assess the cellular internalization capability of LNM nanoparticles compared to free manganese ions, we utilized MC38 cells as a model. As illustrated in Figure 3A, the intracellular manganese ion content in the LNM group was significantly higher than that in the free manganese ion group. We further investigated the cellular uptake capacity of LNM using flow cytometry. As presented in Figure 3B, LNM nanoparticles commenced cellular uptake as early as 2 h, with intracellular fluorescence gradually intensifying over time, signifying the enhanced endocytosis of LNM. Notably, intracellular fluorescence decreased at 10 h, suggesting that the LNM nanoparticles had dissolved within the cells.

LNM nanoparticles have the capacity to dissolve within cells and release free Mn^2+^ ions. These free Mn ions can subsequently activate the cGAS-STING pathway, thereby influencing a cascade of downstream immune-related events. These events include stimulating the activation of the cGAS-STING pathway, as well as the self-phosphorylation, recruitment, and phosphorylation of interferon regulatory factor 3 (IRF3), etc. [5]. To evaluate the activation potential of different concentrations of LNM on the cGAS-STING pathway, we employed an analysis and quantified target protein levels through a gray scale analysis. As shown in Figure 3C–F, we found that LNM nanoparticles can effectively activate the cGAS-STING pathway at a concentration of 10 μM, while LNM nanoparticles have a significant activation effect on the cGAS-STING signaling pathway at a concentration of 50 μM.

### 3.4. Anti-Tumor Effect of LNM Nanoparticles

In our experiments, we confirmed the immune-activating properties of LNM nanoparticles. To extend our investigations to any setting, we examined the role of LNM nanoparticles. Initially, we assessed the distribution of LNM nanoparticles in mice, as illustrated in Figure 4A,B. The results indicate that, in comparison to free DiR, the fluorescence signal of the LNM-DiR group at the tumor site exhibited significant enhancement, with LNM-DiR reaching the tumor site by the 8 h time point. Additionally, the organ fluorescence distribution map reveals that, unlike free DiR, LNM is predominantly enriched in tumors and the spleen. There is still a portion of nanoparticles enriched in the liver, which is consistent with the biodistribution of normal nanoparticles [16]. And the spleen is a crucial immune organ in the body [17]. The spleen contains a large number of lymphocytes and macrophages and is a key site for T cell activation and B cell differentiation into antibody-producing splenic plasma cells (SPPCs) [18]. It plays an important role in anti-tumor immunity. The enrichment of LNM in the spleen can effectively activate host immunity and thus play an anti-tumor role [19]. Subsequently, we investigated the anti-tumor effect of LNM nanoparticles in the MC38-loaded tumor mouse model [20]. As shown in Figure 4C–E, the tumor volume of control mice was obviously the largest by a significant margin, and the free manganese ion group slightly inhibited tumor growth, while the treatment with LNM significantly inhibited tumor development.

### 3.5. Immunoactivation Effect of LNM

LNM nanoparticles act as immune activators and play a critical role in anti-tumor immunity by stimulating the host’s immune response [21]. Therefore, we conducted a further examination of the immune status in the mice. T lymphocytes play an important role in the immune response. As helper T cells, CD4^+^ T cells play an important role in cell-mediated immunity [22]. CD4^+^ T cells mainly recognize foreign antigens presented by antigen-presenting cells and APC, and then regulate other immune cells [23], similar to the activity of B cells or CD8^+^ T cells. CD8^+^ T cells usually differentiate into cytotoxic T cells (CTL) after activation, which can specifically kill target cells directly, and the expression of CD8^+^ T cells is closely related to this therapeutic effect [24]. We first investigated the effect of LNM on the differentiation of mouse spleen T cells in vivo by flow cytometry [25]. As shown in Figure 5A–C, LNM nanoparticles can significantly activate CD4^+^ T cells and CD8^+^ T cells, enhancing the host’s clearance of target cells.

In addition to T cell immunity, there is evidence that some cytokines, especially those involved in adaptive immune responses (e.g., IFN-γ, IL-6) contribute to anti-tumor immunity [26,27,28]. We used a kit to detect the secretion levels of IFN-γ and IL-6 in mouse serum. As shown in Figure 5D,E, the secretion levels of IFN-γ and IL-6 in the LNM group were significantly higher than those in other groups. These results all indicate that LNM can effectively activate host immunity in vivo and play an anti-tumor role.

### 3.6. Safety Analysis of LNM

Considering the good prospects of LNM conversion and application, its safety is one of the most important factors to investigate. We comprehensively investigated and evaluated the safety of LNM nanoparticles. We conducted in vitro experiments to investigate the cytotoxicity of LNM nanoparticles using two normal cell lines. As shown in Figure 6A, LNM nanoparticles did not show significant cytotoxicity in mouse embryonic fibroblasts (3T3) and human embryonic fibroblasts (HEF), and the lethal concentration (IC_50_) of half of the cells in both types of cells was above 100 μM in both types of cells.

Next, we investigated the safety of LNM nanoparticles in vivo. Firstly, drug-treated mice showed a slight increase in body weight during the treatment period, indicating that the LNM nanoparticles are safe for mice (Figure 6B). As illustrated in Figure 6C–F, there were no significant differences in liver-function-related indicators (AST, ALT, ALP) and renal-function-related indicators (CREA-J, UREA, UA) between the treatment group and the blank control group. Finally, we also analyzed tissue sections and performed HE staining on important organs of mice (heart, liver, spleen, lung, and kidney). As shown in Figure 6G, no significant pathological changes were found in the organs of the treatment group.

## 4. Discussion

Microfluidic technology offers a promising approach for producing uniform, stable, and quality-controlled lipid nanoparticles. In this study, we harnessed microfluidic technology to screen and optimize the production process of manganese lipid nanoparticles (LNM). A critical step in LNM preparation is the generation of manganese-containing nanoprecipitates with consistent particle size and stability. The addition of DOPA plays a pivotal role, and these manganese-containing nanoprecipitates can be crafted systematically using microfluidic control. In their research, Sedighi et al. identified FRR as a major factor influencing the diameter and size distribution of fluid dynamics, while TFR significantly impacted productivity [29]. In our quest to select optimal LNM microprecipitated cores, we observed that as the flow rate ratio increased, so did the size of the LNM microprecipitated cores. In other words, when the flow rate ratio was too small, the LNM microprecipitated cores were uniform but too small to be coated with lipids during subsequent processes. This is because film curvature is a crucial factor in the subsequent coating stage. Conversely, when the velocity ratio was too large, the generated metal-ion oil-in-water droplets and phosphoric acid phase droplets generated were too large, making them prone to breakage and resulting in uneven core particle size and poor repeatability. We determined that FRR = 0.2 provided the best flow rate ratio. When combined with the core size and uniformity of LNM microprecipitates observed under an electron microscope, this flow rate ratio yielded core sizes ranging from 30 nm to 60 nm. This choice ensured both the appropriate size of liposomes and robust inter-batch reproducibility of LNM, thus laying the groundwork for potential clinical applications.

In addition, manganese ions have been shown to inhibit tumor development by activating the cGAS-STING pathway, which in turn activates immunity. Recently, Zhao Z et al. demonstrated that Mn^2+^ functions as a potent cGAS activator independent of dsDNA [4]. It may not be surprising that Mn and cGAS-STING are involved in regulating adaptive immunity, given that the cGAS-STING pathway is important in the immunosurveillance of all aberrant cells, including infected, damaged, senesced, mutated, or dead cells [3,30], and the fact that Mn^2+^ greatly sensitizes cGAS-STING and/or directly activates cGAS [4,31]. However, in previous studies, manganese has typically functioned primarily as an immune adjuvant, often necessitating the use of anti-tumor chemotherapy drugs or other immune activators to achieve the desired anti-tumor effect. For instance, Gao et al.’s study indicated that DNA-activated cGAS employs magnesium (Mg^2+^) as its catalytic cofactor. This cofactor catalyzes the conversion of ATP and GTP into 2′3′-cGAMP, forming 2′5′- and 3′5′-phosphodiester bonds [32]. In the current study, both data strongly suggest that LNM can induce CD8^+^ T cell activation in live mice by stimulating the cGAS-STING pathway, leading to the production of type I interferon. This, in turn, promotes the production of pro-inflammatory factors such as IL-6, thereby exerting a significant anti-tumor effect. Notably, this anti-tumor effect can be achieved without the need for combining chemotherapy drugs or other immune activators. LNM nanoparticles alone achieve a tumor inhibition rate of more than 60%. This remarkable tumor inhibition efficiency is likely attributed to the high manganese loading capacity of LNM nanoparticles and their robust manganese utilization.

LNM has several unique characteristics, indicating that it is a promising preparation. The good preparation activity of LNM lies in its good stability after repeated freezing and thawing, and its (1) ability to activate the cGAS-STING pathway and CD8^+^ T cells and promote anti-tumor immune response; (2) ability to induce the production of relevant cytokines such as IFN-γ and IL-6; and (3) use as an immune activating agent but also as a delivery system. No significant side effects were detected.

Manganese ions have demonstrated their effectiveness in numerous scientific studies and clinical applications. Smialowicz et al. demonstrated that intraperitoneal injection of MnCl_2_ can enhance the activity of natural killer cells in mice by mediating increased production of type I interferon (IFN) [33]. However, their practical use is limited due to challenges related to their non-specific distribution in the body and the complexity of their preparation processes. The application of microfluidic technology to develop manganese nanoparticles has addressed these issues. This innovative approach has successfully resolved the problem of free manganese ions, while also ensuring precise control over the stability, uniformity, and reproducibility of manganese nanoparticle preparation. Furthermore, the simplicity and stability of microfluidic technology, combined with the cost-effectiveness and widespread availability of manganese, enhance the promise and feasibility of this preparation method.

## 5. Conclusions

In this study, we developed a quantifiable production method for manganese-based nanoparticles (LNM) utilizing microfluidic technology. Using extensive process optimization and screening processes, we developed a user-friendly, scalable, and efficient technology. The resulting nanoparticles exhibit exceptional stability, uniformity, consistent quality across batches, and a high manganese loading capacity. These LNM nanoparticles, prepared using this method, have demonstrated the ability to effectively activate the cGAS-STING pathway in experiments, enhancing the host’s immune response. Our experiments have further revealed that LNM can induce the activation of CD8^+^ T cells, thereby stimulating the host’s immune response and inhibiting tumor development. Furthermore, LNM exhibits a high level of safety and biocompatibility. The microfluidic screening strategy for lipid manganese nanoparticles not only paves the way for large-scale LNM production but also streamlines the screening process for various nanoparticles in clinical development. This approach opens up new avenues for the widespread application of advanced technologies in the formulation of conventional drug delivery systems.

## Figures and Tables

**Figure 1 pharmaceutics-16-00556-f001:**
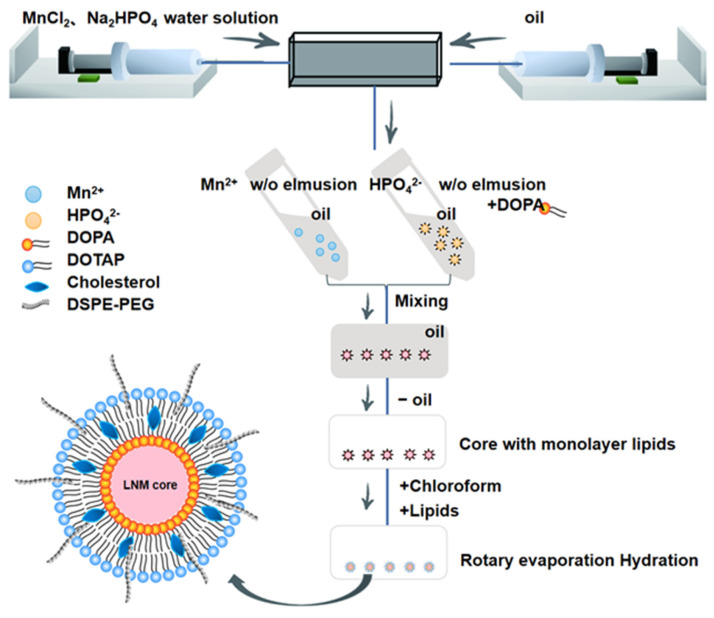
The schematic diagram depicts the preparation process of this LNM nanoparticle. MnCl_2_ solution or Na_2_HPO_4_ solution and CO520/cyclohexane mixed oil phase were introduced into the two side channels of the microfluidic chip, respectively, to form MN^2+^ and HPO_4_^2−^ microemulsion by high-speed shear, and then, the anionic lipid DOPA was introduced to mix the two oil phases, and the LNM core is formed. After that, the oil phase was removed by high-speed centrifugation, the LNM core was collected, and then, some phospholipids (DOTAP/Cholesterol/DSPE-PEG 2000) were added to prepare homogeneous LNM liposomes by film hydration.

**Figure 2 pharmaceutics-16-00556-f002:**
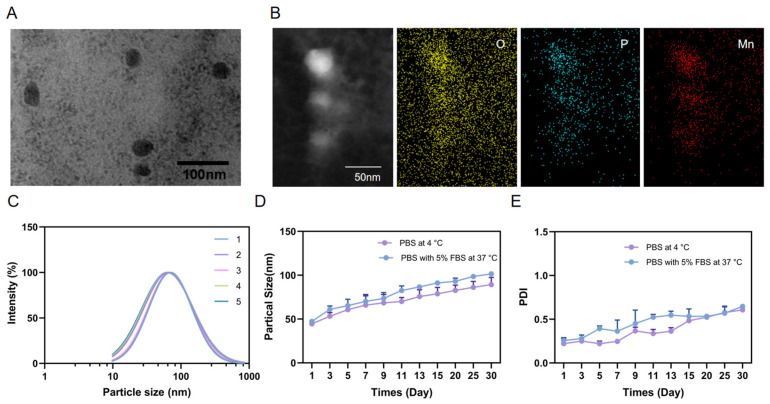
In vitro characterization of LNM. (**A**) TEM of LNM; (**B**) Elemental mapping of LNM; (**C**) Size of LNM prepared in different batches; (**D**) Stability of LNM in PBS at 4 °C and 37 °C in 5% FBS for one month; (**E**) The PDI of LNM for one month in PBS at 4 °C and 37 °C in 5% FBS for one month.

**Figure 3 pharmaceutics-16-00556-f003:**
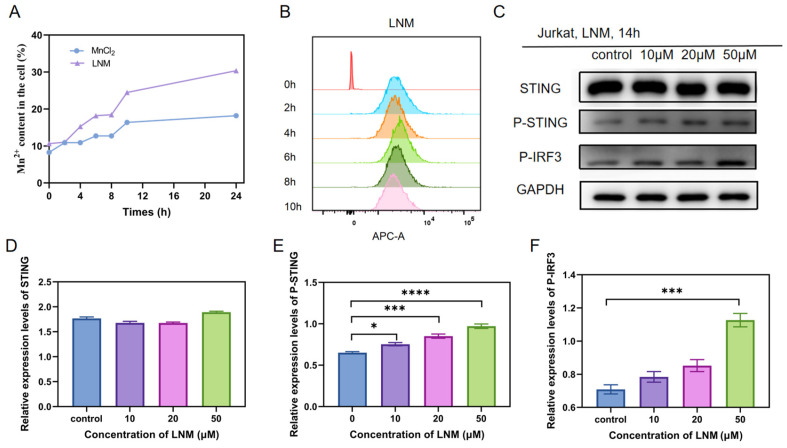
Verification of cells uptake and activation of immunity by LNM (**A**) MC38 cells were incubated with LNM and MnCl_2_, and the content of Mn in MC38 cells was measured by ICP-MS at different time; (**B**) The fluorescence flow cytometry of LNM-Cy5 was measured at different time points; (**C**) Different concentrations of LNM were co-included with Jurkat cells for 14 h, and the activation of cGAS-STING pathway associated proteins was analyzed by Western blot with the indicated antibiotics; (**D**) The expression of STING protein was analyzed by gray scale analysis of WB exposure bands and different concentrations of LNM; (**E**) The expression of P-STING protein was analyzed by gray scale analysis of WB exposure bands and different concentrations of LNM; (**F**) The expression of P-IRF3 protein was analyzed by gray scale analysis of WB exposure bands and different concentrations of LNM. (* *p* < 0.05, *** *p* < 0.001, **** *p* < 0.0001, *n* = 3/group).

**Figure 4 pharmaceutics-16-00556-f004:**
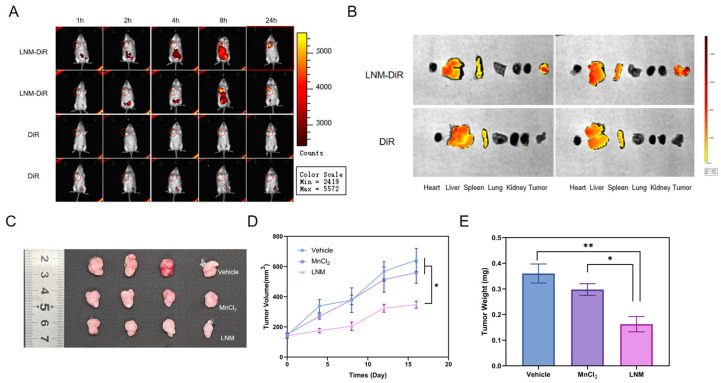
In vivo distribution and anti-tumor effects of LNM nanoparticles. (**A**) Real-time fluorescence imaging of mice within 24 h after intraperitoneal l injection of LNM-DiR and free DiR, and (**B**) fluorescence imaging of major organs in mice 24 h post-injection; (**C**) C57BL/6 mice were inoculated with MC38 cells and LNM (Mn^2+^ = 8 mg/kg) were injected intraperitoneally every 2 days when tumor volume reached 100 mm^3^, (**D**) tumor volume was measured every 4 days and (**E**) tumor weight (* *p* < 0.05, ** *p* < 0.01, *n* = 4/group).

**Figure 5 pharmaceutics-16-00556-f005:**
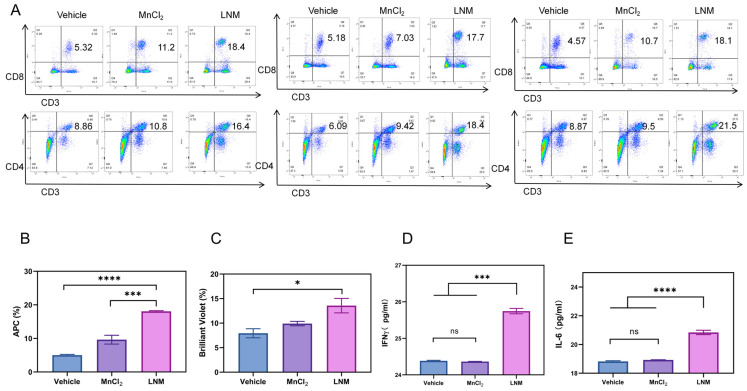
Immunoactivation effect of LNM in vivo. (**A**) Three groups of CD3, CD4 and CD8 T cells from spleen cells of mice treated with different preparations were analyzed by flow cytometry. (**B**) Statistical analysis of CD8 T cell positive rate of mouse splenocytes (*n* = 3/group). (**C**) Statistical analysis of CD4 T cell positive rate of mouse splenocytes (*n* = 3/group). (**D**) The expression of IFN-γ in serum was measured by Elisa (*n* = 4/group), and (**E**) the expression of IL-6 in serum was measured by Elisa Kit (*n* = 4/group). (ns, no significance; * *p* < 0.05, *** *p* < 0.001, **** *p* < 0.0001).

**Figure 6 pharmaceutics-16-00556-f006:**
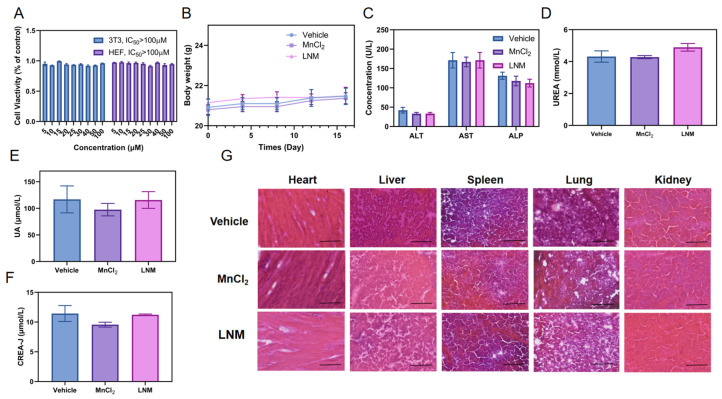
In vivo and in vitro safety analysis of LNM nanoparticles. (**A**) CCK-8 analysis of 3T3 and HEF cell viabilities after 24 h incubated with LNM; (**B**) Changes in body weight of mice after 14 days of administration (*n* = 3); (**C**) Changes in liver function indexes (AST, ALT, ALP) in blood biochemistry of mice after treatment with different formulations for 14 days; (**D**) Changes in blood biochemical renal function indexes (UREA) in mice after 14 days of administration of different preparations; (**E**) Changes in blood biochemical renal function indexes (UA) in mice after 14 days of administration of different preparations; (**F**) Changes in blood biochemical renal function indexes (CREA-J) in mice after 14 days of administration of different preparations; (**G**) H&E staining of mouse vital organs (heart, liver, spleen, lung, and kidney) 14 days after treatment with different formulations.

**Table 1 pharmaceutics-16-00556-t001:** The effect of aqueous and oil phase flow rates on LNM particle size.

Sample	Flow Rate of MnCl_2_ (μL/min)	Flow Rate of Na_2_HPO_4_ (μL/min)	Flow Rate of Cyclohexane/CO-520 (mL/min)	FRR *	Particle Size (nm)
1	42	42	2.1	0.02	<10 nm
2	100	100	2.1	0.048	
3	180	180	2.1	0.086	
4	210	210	2.1	0.1	
5	220	220	2.1	0.105	
6	250	250	2.1	0.119	
7	270	270	2.1	0.129	10–20 nm
8	320	320	2.1	0.152	
9	350	350	2.1	0.167	
10	378	378	2.1	0.18	
11	420	420	2.1	0.2	30–50 nm
12	438	438	2.1	0.209	
13	450	450	2.1	0.214	
14	470	470	2.1	0.223	
15	490	490	2.1	0.233	
16	520	520	2.1	0.248	
17	530	530	2.1	0.252	
18	550	550	2.1	0.262	
19	570	570	2.1	0.271	>50 nm
20	590	590	2.1	0.281	
21	600	600	2.1	0.286	
22	620	620	2.1	0.295	
23	640	640	2.1	0.305	
24	660	660	2.1	0.314	

* FRR: Flow rate ratio.

**Table 2 pharmaceutics-16-00556-t002:** A design of experiences with LNM was conducted to evaluate the impact of varying the volume of DOPA.

Sample	DOPA Conc (mg/mL)	FRR	TFR *	Particle Size (nm)
1	20	0.2	2520	30
2	23	0.2	2520	40
3	26	0.2	2520	20
4	30	0.2	2520	30–50
5	33	0.2	2520	30–50
6	36	0.2	2520	30–50

* TFR: Total flow rate.

**Table 3 pharmaceutics-16-00556-t003:** The samples were analyzed on the day of manufacture, for particle size, and subjected to ICP-MS experiments to measure the content of Mn^2+^ in each milliliter of preparation.

Sample	Volume of LNM Core (μL)	Outer Lipid Content * (mM)	Particle Size (nm)	Mn^2+^ Content (10^5^/mL)	Yield (%)
1	100	2	44.9	11.6	14.06
2	100	5	53.3	12.7	15.39
3	100	10	53.9	34	19.7
4	200	5	56.5	43.8	16.45
5	200	10	55.19	48.4	26.5
6	300	10	53.79	48.8	29.3
7	400	10	45.8	54.3	39
8	500	10	45.2	160.9	41.2
9	600	10	56	239.7	58.11
10	1000	10	47.1	756.1	91.64

* Lipids in formulation (DOTAP: Cholesterol: DSPE-PEG 2000 = 10 mM: 10 mM: 3 mM) concentration refers to the concentration of DOTAP.

## Data Availability

The data presented in this study are available on request from the corresponding author.

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
