# Peer review of "High Manganese Content of Lipid NanoMn (LNM) by Microfluidic Technology for Enhancing Anti-Tumor Immunity"

_pharmaceutics, 2024, doi:10.3390/pharmaceutics16040556_

Round 1
Reviewer 1 Report
Comments and Suggestions for Authors
The manuscript “High manganese content of Lipid NanoMn (LNM) by Microfluidic Technology for enhancing Anti-tumor Immunity” by Sun et al. present the design and development of lipid-based nanoparticle for intra cellular delivery of Mn2+ ions. The authors have performed characterization of their lipid carrier system with in vitro and in vivo efficacy of Mn2+ particles, and bio-compatibility results of their Mn2+ lipid nanoparticles system. While the study has been shown antitumor effect of LNM, the manuscript falls short in providing explanations for the experiments and crucial data to support their claim.
The primary focus of the study revolves around the scale-up manufacturing of lipid nanoparticles containing Mn2+. However, the author does not explain the mechanism behind loading Mn2+ into the lipid core. Mn, in its ionic form is complexed with CO 520 surfactant, which is subsequently mixed with anionic DOPA lipid mixture. Understanding the loading of Mn2+ ions in initial oil phase is challenging. Following the initial complexation, the matrix is mixed lipid layer containing the cationic DOTAP using the well-known lipid hydration method used for liposome production. In reality, the electrostatic interaction between Mn2+ containing DOPA and DOTAP lipid films might lead to the formation of large aggregates, especially since the author does not introduce small drugs or biomolecules. Maintaining a low size and polydispersity index (PDI) of the nanoparticle system without introducing shearing forces during the final complexation step is not explained.
The clearance mechanism for Mn ions from the body remains unaddressed; specifically, it is unclear whether they are excreted through the kidneys. Animals received eight treatments with a dose of 8 mg Mn2+ per kilogram of body weight, resulting in a total dose of 64 mg Mn2+. This dosage is relatively high, raising concerns about tissue residence of metal. Including data on the tissue Mn clearance profile post-treatment will be interesting.
Cationic lipid components are recognized for their potential to elicit immunogenic reactions, leading to changes in biomarkers. To fully support the antitumor immune response of LNM, the author should include blank lipid nanoparticles without Mn2+ as a control group, along with MnCl2.
The rationale behind selecting the dose and treatment schedule for in vitro and in vivo studies needs clarification. Additionally, the effect of increasing the Mn2+ dose should be explored.
Ex vivo fluorescence imaging is employed to demonstrate the tissue uptake of LNM. It would be interesting to utilize ICG for in vivo live imaging, providing insights into tumor uptake, residence time, and the gradual clearance of LMN over time.
Author can improve the experimental section by providing a rationale for the chosen experimental design.
Mn based other agents such as MnO2 are well studied to improve the cancer therapy outcome. Additionally, Mn based contrast agents such as MnCl2 and Mn chelated compound MnDPDP were clinically approved in the past. These agents are used in real human patients. What improvement author system bring, do they think clinically approved MnDPDP can also work as anti cancer agent? There is a lot of literature related to Mn based agents for cancer therapy and imaging. Author can add the literature review of these system in the introduction section of manuscript.
Overall, more detailed experiments are required to support the claims
Comments on the Quality of English Language
No comments
Reviewer 2 Report
Comments and Suggestions for Authors
The manuscript submitted by Sun et al, employed microfluidic technology to design and develop an innovative preparation process, resulting in the creation of a novel manganese lipid nanoparticle (LNM). Authors have demonstrated the ability to activate the cGAS-STING signaling pathway effectively, induce the production of proinflammatory cytokines, and inhibit tumor development. The manuscript is well written, and data presented nicely, however, the following points should be justified before proceeding to the acceptance.
1. Although lipid-based NPs are believed to passively target and accumulate in tumors through the enhanced permeability and retention (EPR) effect, a growing body of research has revealed that lipid-based NPs, without any ligand-modification for active targeting, mainly accumulate in the liver upon systemic administration.a In the current MS, Free-ICG and LNM-ICG have also been accumulated in liver in a great amount as compared to other organs. Had authors noticed any toxicity or adverse effects on liver?
2. Authors are advised to provide the detailed methodology for ICP-MS, SEM, TEM and DLS including instrument model, buffer, and other necessary conditions.
3. There are other studiesb,c,d involving Mn NPs and STING based antitumor properties, authors are encouraged to discuss these studies in the discussion and let the readers know how this study is different with those studies.
References:
a. Sago et al., Cell Mol Bioeng. 2019, 12(5):389-397
b. Wang et al, Immunity, 2018,48(4):675-687.e7.
c. Yu et al, J Mol Cell Biol, 2022;14(6):mjac042.
d. Zhao et al, Cell Rep. 2020,32(7):108053.
Comments on the Quality of English LanguageMinor grammar mistakes need to be corrected
Reviewer 3 Report
Comments and Suggestions for Authors
The authors present an interesting study aimed at the development of a nanosystem for antitumor applications. The manuscript results from an extensive work but there are several points to correct and clarify before its consideration for publication.
1. In general, English language should be revised. Some parts are difficult to follow or/and not well written, as for instance:
a. Line 52 Recent studies have found that demonstrated the successful….
b. Line 54….mixer structures and flow….
c. Line 56 findings should replace assessments.
d. Line 90: To further enhance the process….
2. Liposome preparation is not clear. In addition, there are many other refences that about the procedure that are not included.
3. Table 1 should be located at the results section and the table header should contain more information about the data presented. What is the meaning of FFR*?
4. Size distribution of the liposomes is an important parameter that is missing in all the samples.
5. Figure 2 lacks information. Y axes of fig C should indicate the meaning of the values. Figure caption is poor, and one B needs to be changed for C, C for D and E is missing.
Comments on the Quality of English LanguageIn general, English language should be revised. Some parts are difficult to follow or/and not well written, as for instance:
a. Line 52 Recent studies have found that demonstrated the successful….
b. Line 54….mixer structures and flow….
c. Line 56 findings should replace assessments.
d. Line 90: To further enhance the process….
There are other parts that should be revised.
Round 2
Reviewer 1 Report
Comments and Suggestions for Authors
The authors incorporated sufficient information and new data to support the study findings.
Comments on the Quality of English LanguageI approve the manuscript, although the authors can further improve the quality of English language.
Author Response
Reviewer #1:
I approve the manuscript, although the authors can further improve the quality of English language
Response: Thanks very much for your advice. I have already polished the language throughout the entire text. And I have marked it in blue text in the revised manuscript.

Reviewer 3 Report
Comments and Suggestions for Authors
The authors have changed the original manuscript according to some of the comments of the reviewer. Tables nedd to be improved but in general, the manuscript is suitable for publication.
Author Response
Reviewer #3: The authors have changed the original manuscript according to some of the comments of the reviewer. Tables nedd to be improved but in general, the manuscript is suitable for publication.
Response: Thanks very much for your advice. I revised part of the table and marked it in blue text in the revised manuscript
